# Response Facilitation Induced by Insulin-like Growth Factor-I in the Primary Somatosensory Cortex of Mice Was Reduced in Aging

**DOI:** 10.3390/cells11040717

**Published:** 2022-02-17

**Authors:** Nuria García-Magro, Jonathan A. Zegarra-Valdivia, Sara Troyas-Martinez, Ignacio Torres-Aleman, Angel Nuñez

**Affiliations:** 1Department of Anatomy, Histology and Neurosciences, Universidad Autónoma de Madrid, 28029 Madrid, Spain; nuria.garciam@uam.es (N.G.-M.); adrianzegarra13@gmail.com (J.A.Z.-V.); sara.troyas@estudiante.uam.es (S.T.-M.); 2Facultad de Ciencias de la Salud, Universidad Francisco de Vitoria, Pozuelo de Alarcón, 28223 Madrid, Spain; 3Cajal Institute, Cibernet (CSIC), 28002 Madrid, Spain; ignacio.torres@achucarro.org; 4Universidad Señor de Sipán, Chiclayo 02001, Peru; 5Achucarro Basque Center for Neuroscience, 48940 Leioa, Spain; 6Ikerbasque Foundation for Science, 48009 Bilbao, Spain

**Keywords:** S1 cortex, IGF-I, IGF-I receptors, response facilitation, aging

## Abstract

Aging is accompanied by a decline in cognition that can be due to a lower IGF-I level. We studied response facilitation induced in primary somatosensory (S1) cortical neurons by repetitive stimulation of whiskers in young and old mice. Layer 2/3 and 5/6 neurons were extracellularly recorded in young (≤ 6 months of age) and old (≥ 20 month of age) anesthetized mice. IGF-I injection in S1 cortex (10 nM; 0.2 μL) increased whisker responses in young and old animals. A stimulation train at 8 Hz induced a long-lasting response facilitation in only layer 2/3 neurons of young animals. However, all cortical neurons from young and old animals showed long-lasting response facilitation when IGF-I was applied in the S1 cortex. The reduction in response facilitation in old animals can be due to a reduction in the IGF-I receptors as was indicated by the immunohistochemistry study. Furthermore, a reduction in the performance of a whisker discrimination task was observed in old animals. In conclusion, our findings indicate that there is a reduction in the synaptic plasticity of S1 neurons during aging that can be recovered by IGF-I. Therefore, it opens the possibility of use IGF-I as a therapeutic tool to ameliorate the effects of heathy aging.

## 1. Introduction

It has been demonstrated that the insulin-like growth factor I (IGF-I) can be a potent stimulator of neuronal activity, participating in numerous brain processes (see for review [1,2,3,4,5]). Indeed, IGF-I increases the spontaneous firing rate as well as the response to afferent stimulation in target neurons [6,7,8,9,10,11]. Furthermore, IGF-I injection enhances fast activity in the EEG of mice and non-human primates [12,13]. The functional implications of these changes must be explored because modulation of the excitability of neurons by IGF-I may be involved in synaptic plasticity such as long-term potentiation (LTP), which are critical for learning and memory processes. Indeed, when IGF-I is reduced, LTP is impaired [14,15]. Consequently, deprivation of IGF-I reversibly impairs learning [15,16].

In rodents, tactile discrimination during exploratory behaviors is based on repetitively and rapidly sweeping back and forth whiskers (whisking) across objects or surfaces in repeated rhythmic movements at frequencies between 4 and 12 Hz, scanning their surroundings to obtain tactile information about nearby objects [17,18,19,20]. It is known that trains of repetitive stimulation of whiskers or during whisking induce a long-lasting response facilitation in S1 neurons of rodents [21,22,23,24]. This response facilitation is mediated by the activation of NMDA glutamatergic receptors and increases during periods of EEG activation induced by IGF-I application [6,23]. However, the possible beneficial effects of IGF-I on cortical responses may be altered in pathological situations in which the circulating IGF-I signaling is decreased, such as diabetes or Alzheimer’s disease [2,25].

Aging is a physiological process accompanied by a decline in cognitive performance [26,27]. Both animal and human studies have shown that aging is associated with alterations in synaptic transmission and structural synaptic changes [28,29] and a consistent loss of hippocampal synaptic connections [28,30]. Brain IGF-I and IGF signaling is also reduced during aging [15,31,32,33,34,35]. Basal and IGF-I-induced activation of the brain IGF-I receptor/Akt/GSK3 pathway were markedly reduced in old mice even though they displayed high levels of brain IGF-I receptors [31,32]. Thus, the reduction in IGF-I effects on cortical neurons can be responsible for the decline of cognitive functions during aging. In the present work, we studied whether IGF-I modulates sensory response in the primary somatosensory (S1) cortex of mice and whether this modulation decreases during normal aging. In addition, we studied if the response facilitation induced in cortical neurons by repetitive stimulation of whiskers was affected by IGF-I, in young and old animals. Our findings showed a reduction in the synaptic plasticity of S1 cortical cells during aging. In addition, IGF-I increased cortical responses to whisker stimulation, facilitating synaptic plasticity in young and old mice.

## 2. Materials and Methods

Experiments were performed on C57BL/6J mice (Harlan Laboratories, Spain) of both sexes. Animals were grouped into two age groups (young mice: 3–6 months of age) and (old mice: 20–22 months of age). All experimental groups were sex-balanced. Animals were housed under standard colony conditions with food and water supplied ad libitum and under a 12–12 h light-dark cycle. Animal procedures followed European guidelines (2010/63, European Council Directives) and were approved by the local Bioethics Committee (Government of the Community of Madrid; PROEX: 181.6/21). Efforts were made to minimize animal suffering as well as to reduce the number of animals used.

### 2.1. Recordings and Tactile Stimulation

Animals were anesthetized with isoflurane (2% induction; 1–1.5% maintenance doses) and placed in a David Kopf stereotaxic apparatus (Tujunga, CA, USA). Body temperature was set at 37 °C through a water-heated pad (Gaymar T/Pump, Orchard Park, NY, USA). The sagittal midline of the scalp was sectioned and retracted. A small craniotomy was drilled over the barrel somatosensory (S1) cortex (coordinates from Bregma: 0.5–2 mm posterior; 3–4 mm lateral; according to Paxinos and Franklin, 2003). Unit activity was recorded in the cortical representation of whiskers in the S1 cortex through a tungsten microelectrode (2–5 MΩ, A-M Systems, Sequim, WA, USA). Unit recordings were filtered between 0.3 and 3 kHz and amplified using a DAM50 preamplifier (World Precision Instruments, Sarasota, FL, USA). Signals were sampled at 10 kHz through an analog-to-digital converter (Power 1401 data acquisition unit, Cambridge Electronic Design, Cambridge, UK) and fed into a PC for an off-line analysis with Spike 2 software (Cambridge Electronic Design, Cambridge, UK).

Whisker deflections were evoked by brief air-pulses using a pneumatic pressure pump (Picospritzer, Hollis, NH, USA; 1–2 kg/cm^2^, 20 ms duration), delivered through a 1 mm inner diameter polyethylene tube. All whiskers were first trimmed to a length of 5 mm to avoid complex responses due to multiple whiskers’ deflections. The experimental protocol consisted of air pulses delivered to the whisker at 0.5 Hz during 10 min for basal recording followed by a stimulation train of air pulses at 8 Hz during 10 s to induce response facilitation. After the stimulation train, the whisker was stimulated during 30 min at 0.5 Hz.

### 2.2. Drugs

IGF-I was locally delivered in the S1 cortex (10 nM; 0.2 μL; coordinates as above) employing a 1 μL Hamilton syringe connected to a manual microsyringe injection holder (World Precision Instruments).

### 2.3. Immunohistochemistry

Animals were euthanized with an overdose of pentobarbital (50 mg/kg) and perfused transcardially with saline 0.9% followed by 4% paraformaldehyde in 0.1 N phosphate buffer (PB); pH 7.4. Coronal 50-μm-thick brain sections were cut in a vibratome and collected in PB 0.1 M. Sections were incubated in permeabilization solution (PB 0.1 M, 1%Triton X-100, NHS 10%), followed by 24 h incubation at 4 °C with primary antibodies (1:500) in blocking solution (PB 0.1 M, 1%Triton X-100, NHS 10%). After washing three times in PB, Alexa-coupled goat anti rabbit and goat anti-mouse polyclonal secondary antibodies (1:200, Molecular Probes, Eugene, OR, USA) were used.

Finally, a 1:3000 dilution in PB of Hoechst was added for 5 min. Slices were rinsed several times in PB, mounted with gerbatol mounting medium, and allowed to dry. The omission of the primary antibody was used as a control. The rabbit polyclonal IGF-I Receptor-β antibody (Santa Cruz, CA, USA, 713/AC) and mouse monoclonal anti-NeuN (Abcam, Cambridge, UK, ab177487) were used. The blue-fluorescent DNA stain, DAPI, was also used.

Confocal microscopy images of the somatosensory cortex were obtained with a TCS SP5 Spectral Leica confocal microscope (Leica; Wetzlar, Germany) using a 20× and a 63× oil immersion objective. Image stacks were acquired at 1024 × 1024 pixels using the Leica LAS AF software.

The images obtained were collapsed to create TIFF files with the projections of maximum intensity using a thickness of 10 microns of tissue in all cases. Images were converted to 8-bit grayscale using ImageJ image analysis software for Windows (Microsoft; Albuquerque, NM, USA). The same image processing was applied in all cases, limiting itself to minor adjustments to grayscale and brightness to improve viewing. A densitometric analysis of IGF-1R immunoreactivity was performed and optical density measurements were obtained using ImageJ’s ‘Set Measurement’ routine. The gray values were taken to the Graph Pad Prism 9 software (San Diego, CA, USA) where the histograms were composed, and the corresponding statistical analysis was performed.

### 2.4. Behavioral Experiments

To test the ability of mice to discriminate different textures with their whiskers, we adapted the two-trial Y-maze test as described previously [36,37]. The Y-maze was constructed of black-painted wood with three arms, each 25 cm long, 5 cm wide, and 14 cm high. The walls of the maze arms were covered with two different grades of black sandpaper. While two arms (familiar) were covered with a 500-grit sandpaper, the third (novel) was covered with 220-grit sandpaper. Because the three arms of the maze were identical, and there are no extra-maze cues, the discrimination of novelty vs. familiarity relies only on the different textures that the mouse can perceive with its whiskers. In addition, a similar Y-maze without texture clues was also used (without black sandpaper). The experiments were conducted in a room with dim illumination. During the acquisition phase the mouse was placed at the end of one of the familiar arms (in a random order) and it was allowed to explore both familiar arms for 5 min while the third arm (novel) was closed with a guillotine door. At the end of the first trial the mouse was returned to its home cage for 5 min. In the retrieval phase the mouse was again placed at the end of the same arm where it started the acquisition phase and was allowed to freely explore all three arms for 5 min. To remove any possible olfactory cues, the maze was cleaned with 70% ethanol between the trials. An off-line analysis of the videos was carried out to quantify the time spent exploring the arm that contained the novel texture.

### 2.5. Data Analysis and Statistics

Peristimulus time histograms were used to calculate spike responses in a 50 ms post-stimulus time window following each stimulus (PSTH; 1 ms bin-width). The mean response during the basal recording (10 min) was considered 100%; the response after the stimulation train was calculated each 5 min during the 30 min of recoding. Interval histograms were also calculated to determine the discharge pattern evoked by the whisker stimuli in a 50 ms post-stimulus time window. We selected 3 time periods (1–5, 5–10 and 10–15 ms) and we calculated the percentage of spike intervals in each time period in the different experimental conditions.

A statistical analysis was performed using the Graph Pad Prism 9 software (San Diego, CA, USA). Depending on the number of independent variables, normally distributed data (Kolmogorov-Smirnov normality test), and the experimental groups compared, we used either Student’s *t*-test or two-way ANOVA followed by Sidak’s multiple comparison test. For non-normally distributed data, we used the Mann–Whitney U test to compare two groups. Results are shown as mean ± standard error of the mean (SEM) and *p* values coded as follows: * *p* < 0.05, ** *p* < 0.01, and *** *p* < 0.001.

## 3. Results

### 3.1. Whisker Responses Increased in the Presence of IGF-I in Young and Old Animals

Barrel cortical neurons were recorded in layer 2/3 (up to 500 μm below the pia; supragranular cells) or in layer 5/6 (800–1000 μm; infragranular cells) of the S1 cortex. Neurons were silent or displayed a low firing rate (<2 spikes/s) in spontaneous conditions. All neurons displayed a response to contralateral displacements of one or two whiskers. Whisker responses had on average 1.3 ± 0.13 spikes/stimulus at 20 ± 0.9 ms latency in layer 2/3 neurons (n = 29) from young animals and 1.1 ± 0.14 spikes/stimulus at 18 ± 0.7 ms latency in layer 2/3 neurons from old animals (n = 13; Figure 1A). No significant differences (*p* > 0.05) were observed in the number of spikes or in the response latency between both animal groups. Whisker responses in layer 5/6 neurons had on average 1.2 ± 0.18 spikes/stimulus at 19 ± 0.6 ms in young animals (n = 14) and 1.4 ± 0.21 spikes/stimulus at 18 ± 0.5 ms in old ones (n = 11; Figure 1C). Similarly, no significant differences were found in the number of spikes and the response latency in both animal groups. The low spontaneous firing rate and the reduced tactile responses to whisker stimulation provide strong support that recordings were obtained from pyramidal cells in the barrel cortex, as was reported previously [6,38,39,40,41].

It is known that IGF-I increases neuronal response and its neuronal response is reduced in aging (see Introduction). Thus, we studied its effect on whisker responses in young and old mice. In agreement with previous findings, local application of IGF-I in the S1 cortex (10 nM; 0.2 μL) increased whisker responses 128% in layer 2/3 neurons of young animals (3.1 ± 0.4 spikes/stimulus, 15 min after IGF-I cortical application; n = 13; *p* = 0.002 with respect to basal values) and 89% in layer 2/3 neurons of old animals (2.3 ± 0.4 spikes/stimulus, 15 min after IGF-I cortical application; n = 11; *p* = 0.018 with respect to basal values; Figure 1A). Equally, IGF-I increased 74% the response of layer 5/6 neurons of young animals (2.8 ± 0.4 spikes/stimulus; n = 17; *p* = 0.03 with respect to basal values) and 73.9% in layer 5/6 neurons of old animals (2.7 ± 0.3 spikes/stimulus; n = 10; *p* = 0.0006; Figure 1C).

To know if the discharge pattern evoked by whisker stimuli was similar in young or old mice, we calculated the interval histogram of spikes elicited in a 50 ms time window after the stimulus onset. We selected three ranges of intervals (1–5, 5–10, and 10–15 ms) and we calculated the percentage of spike intervals in each range (Figure 1B,D). In young animals, most spike intervals lasted between 1 and 5 ms. These percentages remained equal in old animals because differences were not statistically significant (Figure 1B,D). However, we found a statistically significant reduction in the percentage of long intervals (10–15 ms) between layer 2/3 neurons of young and old animals (9.7 ± 2.4% vs. 3.0 ± 0.9%; *p* = 0.021). This difference was due to a short-lasting response in old animals. Direct application of IGF-I in the S1 cortex increased the number of spikes elicited by the stimulus and, in addition, significantly increased the percentage of long spike intervals in old animals, reaching the values observed in young animals (10.3 ± 3%; *p* = 0.0246 respect to values in old animals without IGF-I application; Figure 1B). No statistically significant differences were observed in neurons recorded in layer 5/6 after IGF-I application (Figure 1D).

### 3.2. Response Facilitation in Young and Old Mice. Effect of IGF-I Cortical Application

Different studies have demonstrated that repetitive stimulation of whiskers increases tactile responses in layers 2/3 and 5/6 for at least 60 min (e.g., Barros-Zulaica et al. 2014). Here, we tested if this long-lasting response facilitation was altered in old mice. After 10 min of basal whisker stimulation at 0.5 Hz (basal condition; 100%), we applied a stimulation train at 8 Hz during 10 sec to mimic the exploratory behavior of mice (whisking; see Introduction). After that, we maintained the 0.5 Hz stimulation up to 30 min (Figure 2A). In young animals, layer 2/3 neurons increased 86.5% their mean response at 30 min after the stimulation train (1.5 ± 0.14 to 2.4 ± 0.32 spikes/stimulus; n = 29; *p* = 0.007, with respect to basal values). In contrast, layer 2/3 neurons from old animals decreased their response by 12% (1.3 ± 0.12 to 1.2 ± 0.26 spikes/stimulus; n = 13; *p* = 0.4, with respect to basal values) 30 min after the stimulation train (Figure 2B,C).

Layer 5/6 neurons did not show response facilitation after the stimulation train in both young and old animals (Figure 2D). Layer 5/6 neurons increased 28.05% their mean response at 30 min after the stimulation train (from 1.78 ± 0.18 to 2.06 ± 0.24 spikes/stimulus; n = 14; *p* = 0.5 with respect to basal values). Layer 5/6 neurons in old animals increased their response 34.4% (from 1.72 ± 0.21 to 2.18 ± 0.5 spikes/stimulus; n = 12; *p* = 0.96 with respect to basal values) 30 min after the stimulation train (Figure 2E).

To determine the number of cells that were facilitated by the 8 Hz stimulation train in young and old animals, we considered a response facilitation when the neuron increased its response by more than 10%, measured at 30 min after the stimulation train. In young animals, the 8 Hz stimulation train induced a response facilitation in 58.6% of layer 2/3 neurons (17 out of 29 neurons) while in old animals only 23.1% of layer 2/3 neurons (3 out of 13 neurons) showed response facilitation (Figure 2F). Similarly, 50% of layer 5/6 neurons (7 out of 14 neurons) showed response facilitation 30 min after the stimulation train in young animals, while only 36.4% of layer 5/6 neurons in old animals, showed response facilitation (4 out of 11 neurons; Figure 2F). Pooling together these data indicated that synaptic plasticity was diminished in old animals.

In the presence of IGF-I, the 8 Hz stimulation train increased the percentage of neurons that showed response facilitation in both young and old animals. In young animals, the percentage of layer 2/3 neurons that showed response facilitation increased from 58.6% in control conditions (without the application of IGF-I; see above) to 100% (15/15) when the 8 Hz stimulation train was applied 15 min after IGF-I cortical application (Figure 2F). In layer 5/6 neurons of young animals, the percentage increased from 50% (7 out of 14 neurons) in control conditions to 83.3% (15/18) after IGF-I application (Figure 2F). This facilitatory effect of IGF-I was also more evident in old animals. The number of neurons that were facilitated by the stimulation train increased in layer 2/3 from 23.1% (3 out of 13 neurons; see above) to 100% (11 out of 11). Equally, the percentage of layer 5/6 neurons increased from 36.4% (4 out of 11) in control conditions to 100% (13 out of 13) in the presence of IGF-I.

The response facilitation evoked by the 8 Hz stimulation train observed in the presence of IGF-I in young animals (n = 8) was strongly increased. Plots in Figure 3 shows the response to whisker stimulation in control conditions (without the addition of IGF-I; data shown in Figure 2A,C) and data from neurons that were stimulated at 8 Hz 15 min after IGF-I cortical application. Layer 2/3 neurons of young mice showed a response facilitation of 295% at 30 min (from 1.5 ± 0.11 to 4.4 ± 0.11 spikes/stimulus; *p* < 0.0001 with respect to basal values; n = 15; Figure 3A). The time course of the response facilitation was statistically significant between neurons recorded in control conditions and those recorded in the presence of IGF-I (Two-Way Repeated Measure ANOVA, group factor: F (1, 128) = 139.5; *p* < 0.001). In old animals (n = 6), layer 2/3 neurons recorded in the presence of IGF-I changed their response pattern to the 8 Hz stimulation train from a 12% response depression observed in control (see Figure 2B) to a response facilitation of 187% (from 1.3 ± 0.11 to 2.6 ± 0.7 spikes/stimulus; *p =* 0.013 with respect to basal values; n = 11; Figure 3A). Accordingly, the time course of the response facilitation was statistically significant between neurons recorded in control conditions and those recorded in the presence of IGF-I, in old animals (Two-Way Repeated Measure ANOVA, group *<* factor: F (1, 154) = 80, 89, *p* < 0.001). However, the response facilitation observed in presence of IGF-I was significantly lower in old animals than that found in young animals (Two-Way Repeated Measure ANOVA, group factor: F (1, 161) = 12.48; *p* = 0.0005).

Neurons recorded in layer 5/6 in young animals (n = 14) showed 18% of response increment at 30 min after the stimulation train (see Figure 2D). However, the response increased up to 19.3% when the 8 Hz stimulation train was applied in the presence of IGF-I (from 2 ± 0.24 to 3.7 ± 0.57 spikes/stimulus; *p* = 0.029 with respect to basal values; n = 15; Figure 3B). The time course of both responses was statistically different (Two-Way Repeated Measure ANOVA, group factor: F (1, 189) = 58.75, *p* < 0.001). A similar effect was found in neurons of layer 5/6 recorded in old animals. In the control condition the 8 Hz stimulation train induced a 31% response increment (n = 12; see Figure 2D); however, this increment was enhanced to 24.2% (from 1.2 ± 0.10 to 2.6 ± 0.5 spikes/stimulus; *p* = 0.0017 with respect to basal values; n = 12; Figure 3B) when the stimulation train was applied in the presence of IGF-I. The time course of both responses was also statistically different (Two-Way Repeated Measure ANOVA, group factor: F (1, 147) = 58.29, *p* < 0.001). Note that no differences were found in the response facilitation observed in young animals compared with old ones in layer 5/6 in the absence of IGF-I (Two-Way Repeated Measure ANOVA, group factor: F (1, 168) = 1.8). However, differences were found between the response facilitation in old and young animals in the presence of IGF1, being higher in the case of old animals (Two-Way Repeated Measure ANOVA, group factor: F (1,189) = 7.21, *p* = 0.0079).

### 3.3. The Expression of IGR-IR Was Reduced in the S1 Cortex of Old Mice

The expression of IGF-IR was present in all layers of the S1 cortex of young and old animals (Figure 4A,B). Young animals showed a higher expression of IGF-IR in layer 5/6 compared with layer 2/3 (n = 6; *p* = 0.01). However, a clear reduction in IGF-IR was observed of all cortical layers of old compared with young animals (layer 2/3 young vs. old n = 6; *p* = 0.018 and layer 5/6 young vs. old n = 6; *p* = 0.0016). This decrease in the expression of IGF-IR was observed mainly in neurons of superficial and deep layers of the S1 cortex. (Figure 4E,F).

### 3.4. The Performance of a Whisker Discrimination Task Is Impaired in Old Mice

Furthermore, we evaluated the ability of the mice to discriminate the different textures in the Y-maze by comparing groups of young and old mice. We used a test based on the ability of the mice to discriminate different textures in the arms of a Y-maze. When the animals were placed in the apparatus without texture clues, young and old mice behaved similarly (*p* = 0.2589, n = 7; Figure 5A,B), indicating normal ambulatory activity. However, old mice spent less time examining the arm with the new texture compared with the young mice, indicating that old animals have deteriorated texture discrimination (*p* = 0.01; n = 7; Figure 5C,D). Consequently, our results showed that the performance of a whisker discrimination task is impaired with aging. Taken together, these results suggest that the impair of the sensory processing in the S1 cortex that occur during healthy aging can be due to the reduction in IGF-I responses as well as the number of IGF-IR.

## 4. Discussion

Healthy aging is accompanied by a decline in cortical activity and with impairment in cognitive information processing. According to this assumption, the present study shows that the response facilitation evoked by repetitive stimulation of the whiskers was reduced in old mice. However, response facilitation was improved when IGF-I was applied 15 min before repetitive stimulation in old mice, suggesting that the impairment of synaptic plasticity in these animals may be due to a reduction in IGF-I brain activity. These effects were further supported by the reduction in IGF-IR in both supra- (2/3) and infragranular (5/6) layers of the S1 cortex in old animals. The synaptic plasticity impairment observed in old animals was accompanied by a reduction in the performance of a whisker discrimination task. Therefore, the present results suggest that reduced IGF-I activity may hinder information processing in the cortex, explaining the cognitive deficits observed in aging.

It is known that IGF-I increases neuronal activity, participating in numerous brain processes (see for review [1,2,3,4,5]). As shown here, IGF-I increases neuronal responses to whisker stimulation in layers 2/3 and 5/6 of S1 cortex of young and old mice (see also, [6,12]). There were no differences in the whisker responses between young and old animals in basal conditions. However, the increase in the number of evoked spikes per stimulus elicited by IGF-I was larger in layer 2/3 of young mice than in old ones. This difference may be due to a reduction in the IGF-IR in old animals, as is shown in our immunohistochemistry study and in a previous study [35]. This reduction in IGF-IR expression was observed in neurons; however, we cannot discard that may also occur in glia cells.

Although the number of spikes evoked by whisker stimulation was similar in basal conditions in both groups of animals, we observed a change in the whisker-evoked discharge pattern of layer 2/3 neurons in old mice. There was a reduction in the longer spike intervals observed in the whisker responses in layer 2/3 neurons of old mice with respect to young ones, indicating a reduction in the response duration. This reduction was recovered when IGF-I was applied, supporting that the changes observed in cortical responses during aging may be due to a reduction in the IGF-I activity (see also below). Furthermore, the fact that the addition of IGF-I can restore the values obtained in young mice provides a therapeutic value to the IGF-I.

Barros-Zulaica et al. (2014) demonstrated that a brief period of repetitive stimulation of a single whisker at the frequency at which rodents move their whiskers during exploration induced a long-lasting response facilitation in contralateral S1 cortical cells of anesthetized rats that was mediated by the activation of NMDA, glutamatergic receptors (see also [22]). Consequently, these findings suggest that rhythmic patterns of whisker activity can increase the sensory processing, providing a possible mechanism for learning during sensory perception. Here, the 8 Hz stimulation train induced a response facilitation in the majority of layer 2/3 neurons recorded in young mice (58.6%) and the mean whisker response increased 86.5% at 30 min after the stimulation train. In contrast, the stimulation train facilitated only 23.1% of layer 2/3 neurons recorded in old mice and the mean whisker response decreased 12% at 30 min after the stimulation train. No response facilitation was observed in layer 5/6 neurons of both animal groups. The reduction in response facilitation in old mice can be responsible of the reduction in the performance of a whisker discrimination task observed in this animal group. This suggestion is supported by the fact that specific deletion of IGF-IR in cortical astrocytes (IGF-IR^-/-^) also impairs whisker discrimination [37]. Moreover, we demonstrated that IGF-I application in the cortex favors the induction of the response facilitation in both young and old mice. This IGF-I effect was more evident in old animals in which the percentage of neurons that showed response facilitation increased in all cortical layers, reaching up to 100% (see Figure 2F). Note that after the cortical application of IGF-I both neurons of layers 2/3 and 5/6 were able to have a long-lasting response facilitation. Thus, the results suggest that the reduction in IGF-I cortical levels and/or the reduction in IGF-IR in old animals may be responsible of the reduction in synaptic plasticity.

It is known that healthy aging is associated with alterations in synaptic transmission and structural synaptic changes [10,12,28,29]. Furthermore, aging is associated with a reduction in the GH-IGF-I axis activity, resulting in lower serum IGF-I levels [42,43] and impaired brain IGF-I activity [31]. With increasing age, IGF-I levels decline substantially both centrally and peripherally in both humans and rodents [32,44,45,46,47]. Thus, our findings indicate that there is a reduction in the synaptic plasticity in old mice that can be partially recovered increasing IGF-I brain levels. In agreement with this suggestion, we just published that administration of IGF-I for 28 days through Alzet mini-pumps in old animals increases tactile evoked potentials in the somatosensory cortex [12]. It was also demonstrated that IGF-I increases the expression level of NMDA receptors at the hippocampus in aged rats, facilitating LTP induction [48]. In addition, it was indicated that in the hippocampus the mRNA codifying the NR2B subunit increased in young (11 weeks), but not in older (14–16 months), rats after s.c. injection of IGF-I [49]. They also indicated that the ratio of NR2B to NR2A reflects the probability that synaptic plasticity may occur, and that this ratio increased after IGF-I treatment in young and old rats, suggesting that IGF-I treatment favors synaptic plasticity in hippocampal cells, as it also occurs in our present results. Indeed, Pilger et al. [50] reported that L. stagnalis, a pond snail, has age-associated impairment of memory, which can be restored upon IGF-I treatment. Moreover, IGF-I administration can prevent loss of cognitive performance in humans [51]. According to these findings, our previous studies have also indicated that an IGF-I increase induced by exercise improves memory in young mice [52].

## 5. Conclusions

Our findings indicate that there is a reduction in the synaptic plasticity of S1 cortical cells during aging. The fact that some of the deficits in sensory responses observed in old animals were recovered by the application of IGF-I suggests that the origin of these sensory deficits is, at least in part, a decline of IGF-I activity. Therefore, it opens the possibility of using IGF-I as a therapeutic tool to ameliorate the effects of a heathy aging or of pathologies, which are accompanied by a decrease in IGF-I levels.

## Figures and Tables

**Figure 1 cells-11-00717-f001:**
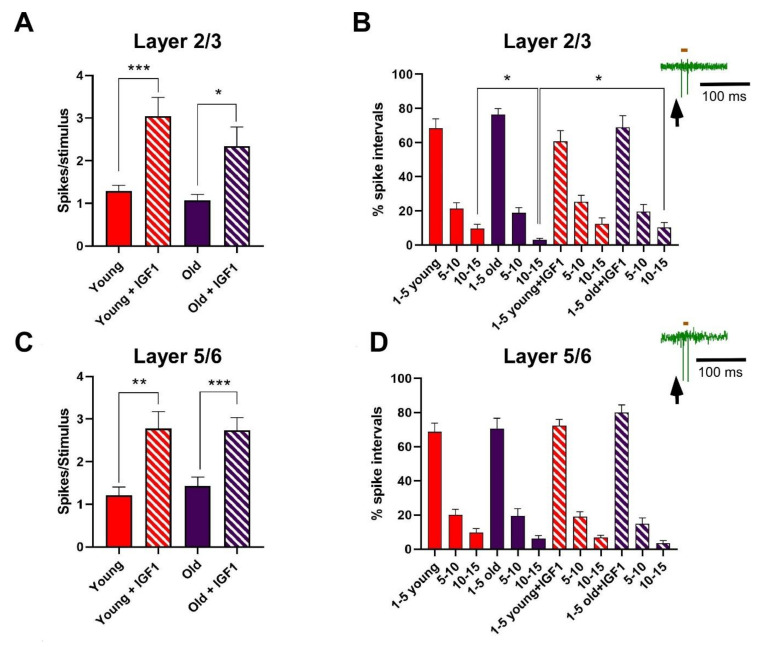
Responses to whisker stimulation increased in the presence of IGF-I in young and old animals in S1 cortex. (**A**) Plot shows the mean number of spikes/stimulus in layer 2/3 neurons in young animals (red) and old (purple) animals in comparison with values obtained 15 min after the application of IGF-I (10 nM; 0.2 μL). Whisker responses were similar in young and old animals; responses increased after IGF-I application in both animal groups. (**B**) The mean percentage of spike intervals in three time ranges (1–5; 5–10 and 10–15 ms) during whisker responses in young and old animals is shown in layer 2/3 neurons. The percentage of spike intervals is shown in control condition and 15 min after application of IGF-I. Data show a significant reduction in the percentage of long intervals (10–15 ms) in neurons recorded from old animals with respect to young ones. The percentage of long intervals in old animals was recovered when whisker responses were recorded after IGF-I application. Inset in B shows a representative response to whisker stimulation in a layer 2/3 neuron; the brown bar on the recording shows a spike interval used to calculate the interval histogram. (**C**,**D**) The mean responses in layer 5/6 neurons as shown in A and B. Note that whisker responses were similar in young and old animals in control conditions and increased after IGF-I application. The percentage of spike intervals did not show significant differences neither between different animal groups nor after the application of IGF-I. All values are expressed as mean ± SEM; Student’s *t*-test; * *p* < 0.05; ** *p* < 0.01, and *** *p* < 0.001 for comparisons.

**Figure 2 cells-11-00717-f002:**
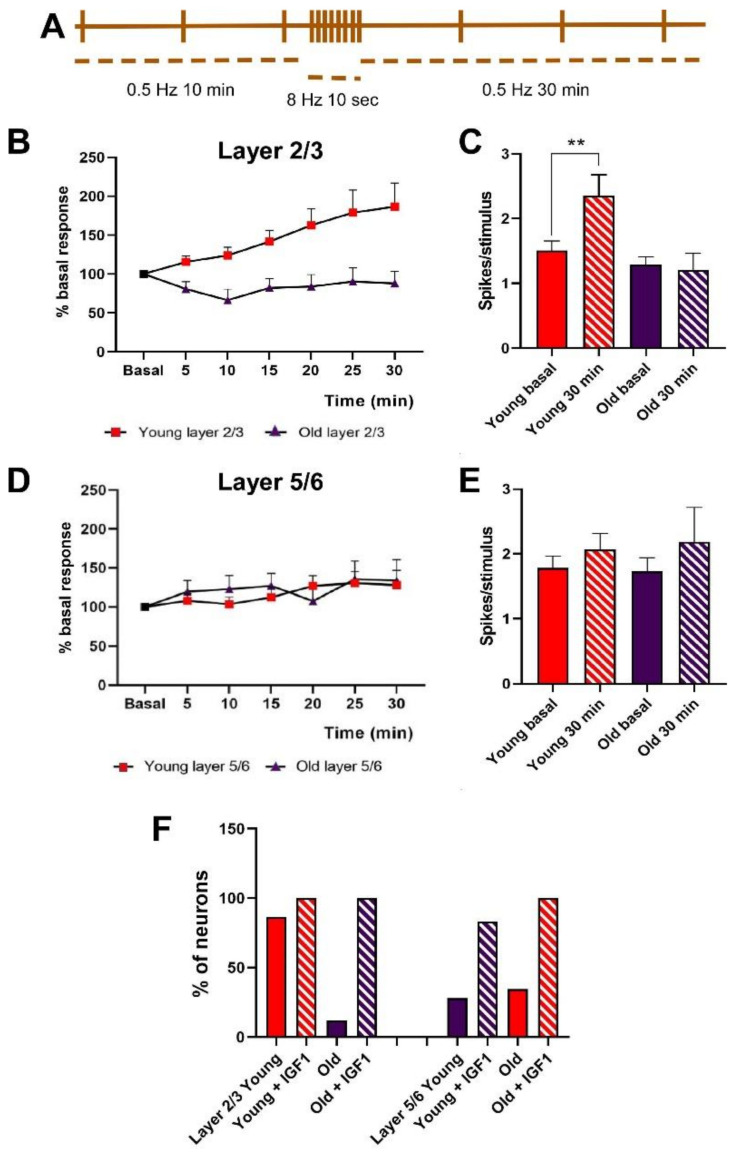
Response facilitation in young and old mice after a stimulation train in the whiskers. (**A**) Stimulation protocol scheme of whiskers to test long-lasting response facilitation. (**B**) Time course recorded in layer 2/3 neurons in young (red boxes) and old (purple boxes) animals. The young animals showed a significant increase in their mean response after the stimulation train (8 Hz; 10 s). The time course was statistically significant compared with old animals that did not increase their response after the stimulation train (two-way ANOVA *F*_(1,280)_ = 27.34 *p* < 0.001). (**C**) Mean responses (spike/stimulus) are shown in baseline conditions and at 30 min after the stimulation train in layer 2/3 neurons (**, *p*< 0.01). Note that only neurons from young animals were facilitated by the stimulation train. (**D**,**E**) Neurons in layer 5/6 did not show a significant increase in their response after the stimulation train in neither young nor old animals. There were also no differences between the two animal groups (two-way ANOVA *F*_(1,161)_ = 0.42 *p* = 0.51). A Student’s *t*-test was used in (**C**,**E**,**F**) The percentage of neurons that showed a facilitation was higher in young animals than in old ones in layer 2/3 of the somatosensory cortex. However, it was similar in layer 5/6 of both groups. The percentage of neurons that showed response facilitation increased in the presence of IGF-I.

**Figure 3 cells-11-00717-f003:**
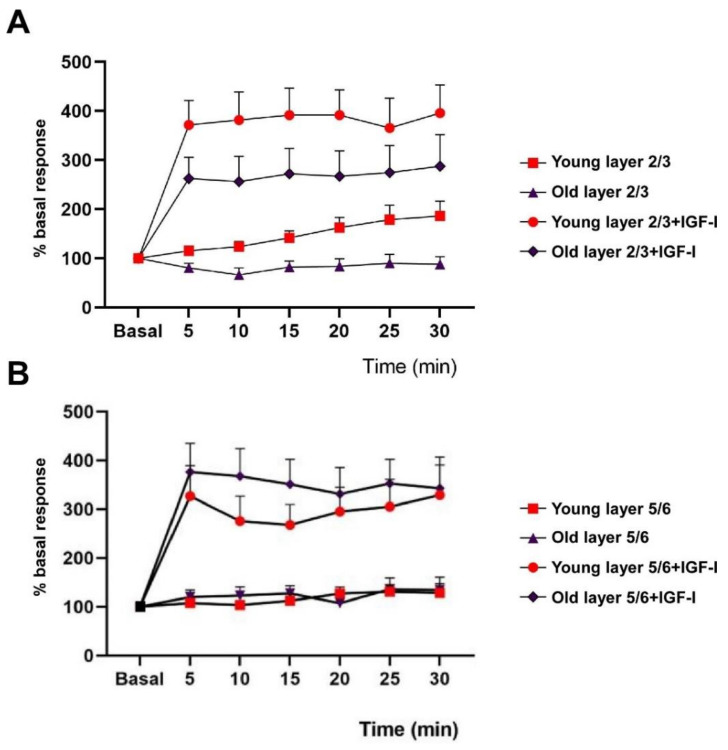
IGF-I increases the response facilitation evoked by the application of the stimulation train. (**A**) In layer 2/3, young mice showed a significant response increase after the stimulation train in the presence of IGF-I (Two-Way Repeated Measure ANOVA, group factor: F _(1, 128, 6)_ = 139.5; *p* < 0.001). A similar effect was found in old animals (Two-Way Repeated Measure ANOVA, group factor: F _(1, 154)_ = 80.89, *p* < 0.001). However, the response increase in the presence of IGF-I was significantly lower in old animals (Two-Way Repeated Measure ANOVA, group factor: F _(1, 161)_ = 12.48; *p* = 0.0005). (**B**) A similar effect was observed in layer 5/6 neurons. In the presence of IGF-I, the response to whisker stimulation increased in young animals (Two-Way Repeated Measure ANOVA, group factor: F _(1, 203)_ = 37.12, *p* < 0.001) and in old ones (Two-Way Repeated Measure ANOVA, group factor: F _(1, 147)_ = 58.29, *p* < 0.001). No differences were found in the response increase in young animals compared with old ones in the presence of IGF-I (Two-Way Repeated Measure ANOVA, group factor: F _(1, 168)_
*p* = 1.8). Data from Figure 2B,D (without IGF-I) are included in these plots for comparison.

**Figure 4 cells-11-00717-f004:**
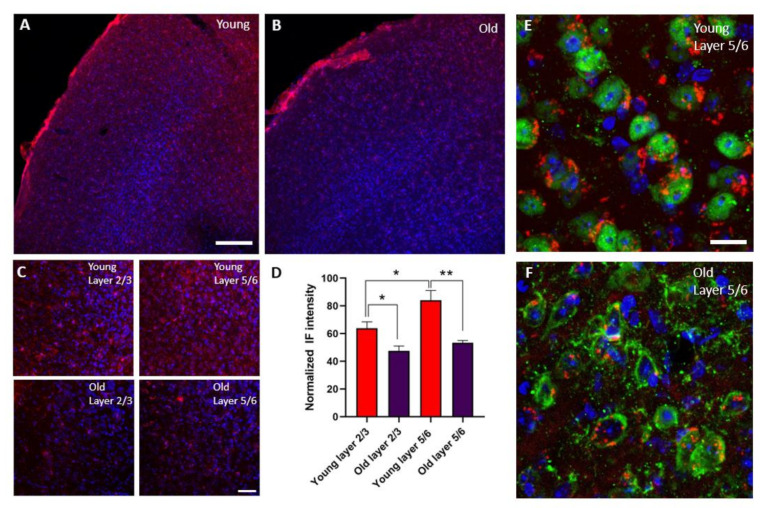
The expression of IGF-1R was reduced in old animals. (**A**,**B**) Panoramic view of double immunocytochemistry in the S1 cortex at low magnification showing staining of IGF-IR (red) and DAPI (blue) in young and old animals. A reduction in IGF-IR expression was observed in all cortical layers in old animals. (**C**) Photomicrographs of double immunocytochemistry in the S1 cortex. Infragranular layer (5/6) showed higher expression of the IGF-IR in young animals. (**D**) Comparisons of densitometric measurements for IGF-1R in layers 2/3 and 5/6 of young and old animals. Values of the optical density are represented as means ± SEM of the normalized fluorescence intensity. Young animals showed a higher expression of IGF-IR in layer 5/6 compared with layer 2/3 (n = 6; *p* = 0.01). A clear reduction in IGF-IR was observed in all cortical layers of old compared with young animals (layer 2/3 young vs. old n= 6; *, *p* = 0.018 and layer 5/6 young vs. old n = 6; **, *p* = 0.0016). (**E**,**F**) Representative high-magnification photomicrographs of neurons located in layer 5/6 showing staining of IGF-IR (red) and NeuN (green) in young and old animals. Most IGF-IR staining was located on cortical neurons and was reduced in old animals. Scale bar in A and B: 120 µm. Scale bar in C: 38 µm. Scale bar in E and F: 15 µm.

**Figure 5 cells-11-00717-f005:**
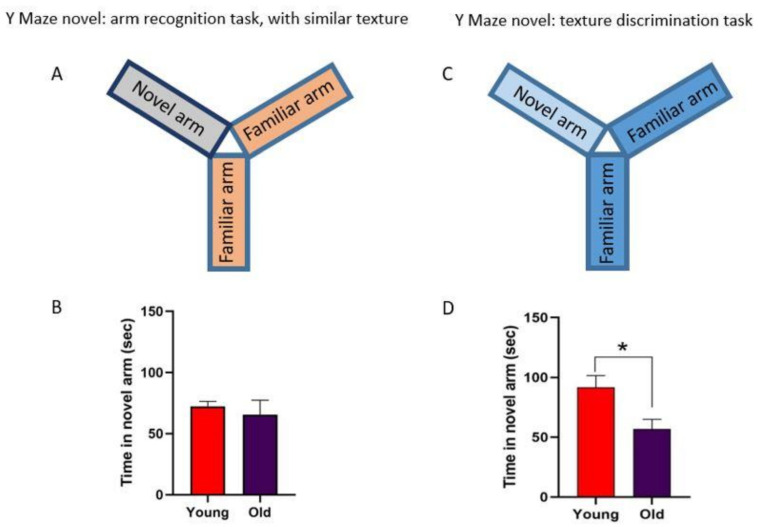
Results of Y-maze whisker-dependent texture discrimination. (**A**,**B**) Diagram and plot showing the time spent in the novel arm without any difference in the texture; young and old mice spend the same time in the novel arm (n = 7 in each group; Mann–Whitney unpaired test *p* = 0.8939). (**C**,**D**) The plot shows the time in seconds that the animal spent on the new arm with the new texture. Young mice spent more time exploring the new arm than old mice (n = 7 in each group; Mann–Whitney unpaired test, two-tailed test *, *p* = 0.012).

## Data Availability

Data are accessible upon personal request.

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
