# Peer review of "Response Facilitation Induced by Insulin-like Growth Factor-I in the Primary Somatosensory Cortex of Mice Was Reduced in Aging"

_cells, 2022, doi:10.3390/cells11040717_

Round 1
Reviewer 1 Report
In the present study, the authors reported that the response facilitation evoked by repetitive stimulation of the whiskers was reduced in old mice, and applying IGF-I in S1 cortex improved the response facilitation. They then reported that the expression of IGF-I receptor (IGF-IR) was reduced in layer 2/3 and layer 5/6 of S1 cortex of old mice versus young mice. Furthermore, they found that the performance of a whisker discrimination task (Y-maze whisker-dependent texture discrimination) was reduced in old mice compared with young mice. Accordingly, the authors believe that the sensory deficits in old mice may be due to the decline of IGF-I activity. The data are interesting. However, I have a number of concerns which I think the authors should address.
- The link between the reduced behavioral performance and the declined activity of IGF-I is not strong enough. This link could be strengthened by comparing the behavioral performance of old mice with or without application of IGF-I in the S1 cortex. Another way is to overexpress the IGF-IR in the S1 cortex of old mice and see if their performance could be improved.
- Images with higher resolution should be provided in Figure 5. It will be helpful to clarify where or in which type(s) of neurons the expression (or reduction of expression) of IGF-IR occurred in the S1 cortex. In terms of the quantification, could you do a western blot analysis to confirm the result of the IGF-IR staining?
- I believe it would be better use one sentence but not two as the title of the manuscript.
Author Response
We like to warmly thank for the important and constructive suggestions that have greatly improved our manuscript. What follows is a point-by-point answer to the reviewer’s suggestions.
- The link between the reduced behavioral performance and the declined activity of IGF-I is not strong enough. This link could be strengthened by comparing the behavioral performance of old mice with or without application of IGF-I in the S1 cortex. Another way is to overexpress the IGF-IR in the S1 cortex of old mice and see if their performance could be improved.
We agree with the reviewer that the suggestion that the decline in behavioral performance may be due to a reduction in IGF-I activity is not proved. However, this suggestion is based on the fact that specific deletion of IGF-IR in cortical astrocytes (IGF-IR-/-) of young mice impairs whisker discrimination, as well (Noriega et al. 2021). In this new version, we indicate in the Discussion section that this suggestion is based on these previous results.
The experiments that you propose are very interesting to prove that the increase in the behavioral performance in old animals is due to IGF-I. We have just published that administration of IGF-I for 28 days through Alzet mini-pumps in old animals increases tactile evoked potentials in the somatosensory cortex (Zegarra-Valdivia et al. 2021), suggesting that this treatment could be favor synaptic plasticity in the cortex. In agreement with this suggestion, it has been demonstrated that IGF-I increases the expression level of NMDA receptors at the hippocampus in aged rats (Sonntag et al., 2000), facilitating LTP induction. However, at this time we do not have a sufficient number of old animals to do this experiment because we would need two groups of at least six animals over 20 months old. We have included in the discussion the above findings, suggesting that synaptic plasticity could be improved with an IGF-I chronic treatment (lines 406-416).
- Images with higher resolution should be provided in Figure 5. It will be helpful to clarify where or in which type(s) of neurons the expression (or reduction of expression) of IGF-IR occurred in the S1 cortex. In terms of the quantification, could you do a western blot analysis to confirm the result of the IGF-IR staining?
We have modified the new Figure 4 to include images with higher magnifications. These new images show that cortical neurons display IGF-I receptor staining, as also occurs in other brain regions such as in orexin neurons (Zegarra-Valdivia et al. 2020) or in cholinergic neurons (Zegarra-Valdivia et al. 2021). However, we cannot rule out that the receptors also decrease in other cell types such as in astrocytes (see Noriega et al. 2021). We have included this issue in the discussion of this new version (lines 369-371).
Thank you very much for your suggestion for quantification. As indicated above, at this time we do not have a sufficient number of old animals to do a Western blot analysis. Taking into account the small size of the S1 cortex of the mouse, we would need to take a slice of cortex from 2-3 control and old mice to have enough sample to do the analysis. In addition, we would need more time than the Journal gives us to do these experiments.
- I believe it would be better use one sentence but not two as the title of the manuscript.
We have changed the title as you suggest.

Reviewer 2 Report
García-Magro et al. present an interesting paper on the potential role of IGF-1 in the cognitive decline associated with aging, that will be of broad interest to the field.
I found the paper well written overall, and understandable in the most part. A small note - is it this Journals policy to include so much numerical data (mean, error margins, p value, statistical test, controls) in parenthesis throughout the results text? While for some core experiments exact numbers/changes can of course be reported, currently the amount of data given within the text, makes the results section very difficult to read. The flow of the text really suffers. Please see where this data can be moved/shorted to absolute necessity.
Additional minor comments:
- Fig 3: please add error bars so that the reader can assess the spread of the data.
- Consider combining Figure 2 and 3
- In the Fig6 experiment, is it not possible to administer IGF-1 somehow to these mice to assess whether the effect can be rescued?
- Line 439: 'ex-periments' should be 'experiments'
Author Response
We like to warmly thank you for the important and constructive suggestions that have greatly improved our manuscript. What follows is a point-by-point answer to the reviewer’s suggestions.
I found the paper well written overall, and understandable in the most part. A small note - is it this Journals policy to include so much numerical data (mean, error margins, p value, statistical test, controls) in parenthesis throughout the results text? While for some core experiments exact numbers/changes can of course be reported, currently the amount of data given within the text makes the results section very difficult to read. The flow of the text really suffers. Please see where this data can be moved/shorted to absolute necessity.
Thank you very much for your suggestions. We have reduced the amount of data in the result section.
Additional minor comments:
- Fig 3: please add error bars so that the reader can assess the spread of the data.
The new figure 2F shows the percentage of neurons that were facilitated in each condition. Thus, bars are not a mean and we cannot include the error bars.
- Consider combining Figure 2 and 3
We have included the old figure 3 in the new Figure 2F as you suggested.
- In the Fig6 experiment, is it not possible to administer IGF-1 somehow to these mice to assess whether the effect can be rescued?
As is indicated above in the answer to the reviewer 1, we have just published that administration of IGF-I for 28 days through Alzet mini-pumps in old animals increases tactile evoked potentials in the somatosensory cortex (Zegarra-Valdivia et al. 2021). The experiments that you propose are very interesting to prove that it is due to IGF-I. However, at this time we do not have a sufficient number of old animals to do this experiment because we would need two groups of at least six animals over 20 months old. Also, we would need more time than the Journal gives us to answer your suggestions.
We have included in the discussion that synaptic plasticity could be improved with an IGF-I chronic treatment (lines 412-416).
- Line 439: 'ex-periments' should be 'experiments'
Thanks, the typo has been corrected.

Round 2
Reviewer 1 Report
In the revised manuscript, the authors reasonably addressed my questions/concerns raised in the previour review. I have no further questions.